

# Multiple Early Holocene eruptions of Katla produced tephra layers with similar composition to the Vedde Ash

David J. Harning[1], Thor Thordarson[2], Áslaug Geirsdóttir[2], Gifford H. Miller[1, 3], Christopher R. Florian[4]

[1]Institute of Arctic and Alpine Research, University of Colorado, Boulder, USA
[2]Faculty of Earth Sciences, University of Iceland, Reykjavík, Iceland
[3]Department of Geological Sciences, University of Colorado, Boulder, USA
[4]National Ecological Observatory Network, Boulder, USA

*Correspondence to*: David J. Harning (david.harning@colorado.edy)

**Abstract.** The Vedde Ash, first described in Norway and dated to ~12000 cal a BP, has been taken to represent tephra derived
from a large eruption of the Katla volcano in Iceland and dispersed across the North Atlantic and Europe. However, evidence
for tephra layers with similar composition to the Vedde Ash, but of different ages, questions the utility of isolated Vedde-like
tephra layers as reliable and independent age control. Here, we report three stratigraphically separated Early Holocene Katla
tephra layers from the lake Torfdalsvatn, in north Iceland, that have bimodal chemical composition similar to the Vedde Ash.
By using previously published conventional [14]C ages and revised calibration curves, we provide new ages for these tephra
layers of ~11375, 11360, and 11200 cal a BP – all substantially younger than the Vedde Ash. Torfdalsvatn's record stands as
an important reminder that repeated explosive eruptions of Iceland's major volcanos during the deglacial cycle have produced
multiple tephra plumes with similar geochemistry that may span 1000s of years. As a result, we urge caution when using
isolated Icelandic tephra layers in distant regions as precise geochronometers without supporting age control.

## 1 Introduction

The Vedde Ash, first described in lake sediments from the Ålesund and Nordfjord regions of western Norway (Mangerud et
al., 1984), has become a widespread marker tephra found throughout the North Atlantic and Europe to regions as far as Slovenia
(Lane et al., 2011) and Arctic Siberia (Haflidason et al., 2018). For distal locations, its recorded composition is generally
rhyolitic, however, some sites, including the type locality in Norway, also contain a substantial mafic component (e.g., Lane
et al., 2012). Bayesian modeling of radiocarbon ages from sediments in Europe place the time of Vedde Ash deposition at
12023 ± 43 cal a BP (Bronk Ramsey et al., 2015), which is statistically indistinguishable to the age determined from layer
counting in Greenland ice cores (12121 ± 114 cal a BP, Rasmussen et al., 2006). Given that the Vedde Ash lies within the
Younger Dryas Stadial (or Greenland Stadial 1, 12900 to 11700 cal a BP, Rasmussen et al., 2006), an abrupt cold event within
the Late Glacial to Holocene transition (Mangerud et al., 1974), studies have leveraged its age and widespread distribution to
synchronize rapid climate change across the North Atlantic at this time (e.g., Bakke et al., 2009; Lane et al., 2013).

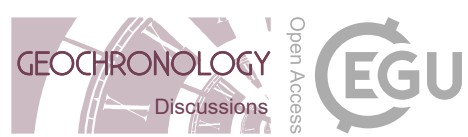

While the Vedde Ash's composition and relatively precise age constraint have historically been relied on for robust age control, other compositionally similar tephra layers from the Katla volcano have now also been identified in the Late Glacial period (e.g., Lane et al., 2012). These include but are not limited to the Dimna Ash (>15100 cal a BP, Koren et al., 2008), the Abernethy Tephra (i.e., AF555, 11790 to 11200 cal a BP, Matthews et al., 2011; Macleod et al., 2015), as well as tephra glass found in the marine realm on the South Iceland Rise that may be up to 3000 years older than the Vedde Ash (Bond

et al., 2001; Thornalley et al., 2011). However, few proximal sites in Iceland have detected these silicic Katla tephra layers, presumably due to substantial ice sheet presence at the time (Norðdahl and Pétursson, 2005). Based on compositional similarities, correlations to the Skógar Tephra in north Iceland (Norðdahl and Haflidason, 1992) and the Sólheimar Ignimbrite adjacent to Katla have been proposed (Tomlinson et al., 2012), but these deposits lack independent age control needed for confirmation. For the latter, the Sólheimar Ignimbrite also lacks the basaltic component found in the type Vedde Ash, and

features a distinct intermediate component (Tomlinson et al., 2012).

Torfdalsvatn, a small lake in north Iceland (66.06°N, 20.38°W, Fig. 1a), contains the longest known, continuous lake sedimentary record in Iceland. Based on conventional radiocarbon ($^{14}$C) ages and tephra layer compositional analysis, Torfdalsvatn's record has been suggested to include the Vedde Ash (Björck et al., 1992; Rundgren, 1995). However, these records precede recent advancements in $^{14}$C calibrations as well as the recognition of multiple silicic Katla tephra layers during

the Late Glacial period. In this paper, we present new and expanded compositional datasets from Torfdalsvatn's earliest sediment, in addition to providing improved age control based on updated radiocarbon calibration curves (Reimer et al., 2020) and Bayesian modeling approaches (Blaauw and Christen, 2011). We argue that contrary to conclusions in earlier studies, Torfdalsvatn does not contain the Vedde Ash. Instead, our results show that three tephra layers in Torfdalsvatn's oldest sediment have similar geochemical composition to the Vedde Ash but were deposited in the following millennia.

## 2 Materials and Methods

### 2.1 Sediment core collection

In February 2012, a continuous lake sediment core was recovered using a Bolivia coring system from a lake-ice platform above the deepest portion of the lake (5.8 m depth, Fig. 1b). This core, TORF12-1A-1B, was collected in 1.5 m increments until reaching deglacial sediment and bedrock at the bottom. For this study, we only focus on the deepest segment that contains

tephra layers deposited shortly after local deglaciation (Fig. 2).

### 2.2 Tephra compositional analysis

Each tephra layer was sampled along the vertical axis, sieved to isolate glass fragments between 125 and 500 μm, and embedded in epoxy plugs. Individual glass shards were analyzed at the University of Iceland on a JEOL JXA-8230 election microprobe using an acceleration voltage of 15 kV, beam current of 10 nA and beam diameter of 10 μm. The international

A99 standard was used to monitor for instrumental drift and maintain consistency between measurements (Table S1). Tephra



origin was then assessed following the systematic procedures outlined in Jennings et al. (2014) and Harning et al. (2018). Briefly, based on $SiO_2$ wt% vs total alkali ($Na_2O+K_2O$) wt%, we determine whether the tephra volcanic source is mafic (tholeiitic or alkalic), intermediate and/or rhyolitic. From here, we objectively discriminate the source volcanic system through a detailed series of bi-elemental plots produced from available compositional data on Icelandic tephra (Fig. S1).

**2.3 Age control**

The TORF12-1A-1B sediment core uses two well-defined marker tephra constraints for primary age control. The first tephra layer (Tv-1) was previously identified in Torfdalsvatn and dated with conventional [14]C ages (Björck et al., 1992). To provide a calibrated age for the Tv-1 tephra layer, we generated a Bayesian age model using the published [14]C dates (Table 1) from Björck et al. (1992) and the recent IntCal20 calibration curve (Reimer et al., 2020). The corresponding depth of Tv-1 in this

age model provides an age of $11460 \pm 560$ cal a BP (Fig. 3a). The second tephra layer used in TORF12-1A-1B's age model is the G10ka Series (formerly Saksunarvatn Ash), which has been geochemically identified in previous Torfdalsvatn records, and is easily identifiable by its >20 cm thickness and relatively coarse-grained nature (Björck et al., 1992; Alsos et al., 2021). However, instead of using the single age from the Greenland ice core (e.g., Alsos et al., 2021), we apply an age of 10400 cal a BP to the lower limit based on the recent recognition that it was generated from multiple eruptions between 10400 and 9900

BP (Harning et al., 2018, 2019; Óladóttir et al., 2020). Age models for the Björck et al. (1992) and TORF12-1A-1B sediment records were both created using the R package rbacon and default settings (Fig. 3, Blaauw and Christen, 2011).

**3 Interpretations**

**3.1 Age model**

Torfdalsvatn's revised and calibrated age model shows a linear sedimentation rate from the base upward, ranging from 11640

to 9740 cal a BP (Fig. 3a). While bulk [14]C in Iceland can be stratigraphically too old due to the introduction of old carbon from the lake catchment (Geirsdóttir et al., 2009), bulk [14]C ages that are stratigraphically too young are unlikely. This aspect is reflected well by the most probable ages be more heavily weighted towards younger [14]C ages in the model and the samples Ua-1889 and Ua-1887 being slightly too old (Fig. 3a). We assume that the difference in bulk [14]C age relative to actual age is also minimal in a recently deglaciated landscape where substantially old carbon storage in soils is not probable due to recent

ice sheet erosion. In this sense, the bulk [14]C ages from Torfdalsvatn should provide reliable ages of sedimentary deposition and firm limiting age constraint for the intercalated tephra layers.

**3.2 Tephra layer descriptions**

Hekla 11460 (12TORF-1B-108): A 0.6 cm thick, black basalt tephra of very fine ash is present at 845 cm depth (Table 1, Fig. 2), and has a modeled age of $11460 \pm 560$ cal a BP (Fig. 3a). The tephra layer is comprised of pristine and vesicular,

sideromelane grains, poorly to non-vesicular black translucent to opaque grains, and grey microcrystalline grains. All 17 grains





analyzed have alkalic basalt composition consistent with the Hekla volcanic system (Fig. 4a and S1; Table S2). The similar homogenous composition and thickness to the lowermost tephra layer as described by Björck et al. (1992) correlate this to the Tv-1 tephra layer (Fig. 4a). The 11460 cal a BP age makes it one of the oldest Hekla tephra so far to be identified in Iceland's terrestrial record and almost 2000 years older than the basaltic Hekla tephra layers (~9600 and 9400 cal a BP) present in
Bæjarvötn lake sediments in northwest Iceland (Fig. 1, Harning et al., 2018).

Katla 11375 (12TORF-1B-100): A 1.1 cm thick, salt and pepper layer of medium ash is located at 837 cm depth (Fig. 2) and has a modeled age of $11375 \pm 330$ cal a BP (Fig. 3b). Visual inspection of the tephra reveals silicic and basaltic grains with delicate protrusions (Fig. 5a). The 41 grains analyzed are all alkalic in their compositions and have chemical attributes consistent with the Katla volcano (Fig. 4b-c and S1; Table S2). These range from basalt (n=20) to low-$SiO_2$ (n=2) and high-
$SiO_2$ intermediate (n=1), to rhyolite (n=18). The similar bimodal compositional range (Fig. 4b) and thickness between Katla 11375 and the Tv-2 tephra described by Björck et al. (1992) supports a correlation.

Katla 11360 (12TORF-1B-99): A 0.9 cm thick, salt and pepper (black and white color) layer of medium ash is located at 835.5 cm depth (Fig. 2) and has a modeled age of $11360 \pm 340$ cal a BP (Fig. 3b). Visual inspection of tephra reveals a mixture of silicic and basaltic grains, where many of the grains feature delicate protrusions (Fig. 5b). All 44 grains analyzed
exhibit alkalic compositions consistent with origin from the Katla system (Fig. 4 and S1; Table S2). These range from basalt (n=19) to low-$SiO_2$ (n=4) and high-$SiO_2$ intermediates (n=2), and rhyolites (n=19). This bimodal compositional range is identical to that of the Katla 11375 tephra below (Fig. 4b). Although not described in Björck et al. (1992), the authors do note two tephra-rich horizons above Tv-2, the equivalent of Katla 11375. The sharp upper and lower contacts between the organic sediment and the 12TORF-1B-99 layer in our core as well as the pristine nature of glass shards suggests that this is a primary
tephra layer and not formed by reworking of the Katla 11375 tephra.

Katla 11200 (12TORF-1B-90): A 0.4 cm thick, black layer of fine to medium ash is located at 827 cm depth (Fig. 2), which has modeled age of $11200 \pm 330$ cal a BP (Fig. 3b). There is approximately a 50/50 split between the tephra vs sediment sub-populations (Fig. 5C), with the latter likely resulting from sampling-induced contamination. Of the 40 grains analyzed, 33 (83%) exhibit alkalic compositions, where the composition of 28 grains indicates origin within the Katla volcanic system,
while five alkali basalt grains have glass composition indicative of the Hekla volcanic system (Fig. 4). The dominant Katla tephra grains range from basalt (n=7) to basaltic icelandite to icelandite (n=3) and rhyolite (n=18). The seven tholeiite basalt tephra grains are consistent with origin from the Kverkfjöll volcanic system (Fig. S1 and Table S2). The dominance of the Katla population and preservation of delicate protrusions (Fig. 5c) is consistent with pristine tephra fall from a Katla eruption. The presence of non-Katla tephra grains may either reflect a concomitant episode of catchment erosion following the initial
deposit of the Katla tephra, or similarly timed eruptions from the Kverkfjöll and Hekla volcanic systems.

## 4 Discussion and Conclusions

Torfdalsvatn's stratigraphic record includes three Late Glacial to Early Holocene tephra layers with similar bimodal Katla chemical compositions (Figs. 2 and 4). Previous lake sediment studies from Torfdalsvatn suggested that the 9 cm of sediment





immediately following the first Katla tephra layer (i.e., Tv-1), which includes two additional unanalyzed tephra layers (Fig.

2), resulted from redeposition of older sediments based on 1) observations of similar pollen spectra compared to a lower level and 2) lack of sediment laminations (Rundgren, 1995). However, the preservation of delicate protrusions on individual glass shards from our three tephra layers (Fig. 5) and organic sedimentation between each layer suggest that each is a primary horizon, and not the consequence of sediment reworking. Collectively, and in conjunction with age model analyses, these lines of evidence suggest that the three Early Holocene Katla tephra layers found in Torfdalsvatn were generated from separate

eruptions between $11375 \pm 330$ and $11200 \pm 330$ cal a BP.

The oldest of these three layers in Torfdalsvatn (formerly Tv-2) has previously been correlated to the Vedde Ash (Björck et al., 1992) and the local Skógar Tephra based on major oxide composition (Norðdahl and Haflidason, 1992). However, the ages of the lowermost Katla tephra layers in our calibration of the Björck et al. (1992) age model ($11270 \pm 420$ cal a BP) and from the TORF12-1A-1B sediment record ($11375 \pm 330$ cal a BP) are inconsistent with the Vedde Ash found in

Europe ($12023 \pm 43$ cal a BP, Bronk Ramsey et al., 2015) and Greenland ($12121 \pm 114$ cal a BP, Rasmussen et al., 2006). In this context, it is noteworthy that near the base of marine sediment core MD99-2269 on the North Icelandic Shelf (Fig. 1a), three rhyolitic Katla cryptotephra with Vedde-like composition are present with indicated ages of $11520 \pm 190$, $10760 \pm 100$, and $10420 \pm 100$ cal a BP (Kristjánsdóttir et al., 2007), adding further support to the notion of three (or more) Katla rhyolite tephra with northerly dispersal. Moreover, the silicic composition of the Vedde Ash in Norway and Scotland are not entirely

consistent with Torfdalsvatn's. For example, FeO wt % for a given $TiO_2$ wt % is lower for the Vedde Ash layers in Europe (Fig. 4d). While the bimodal Skógar Tephra, found ~120 km to the east of Torfdalsvatn in Fnjóskadalur (Fig. 1a), is compositionally similar to the Vedde Ash and the three younger layers in Torfdalsvatn (Fig. 4B-C), there is no independent age control to support either correlation (Norðdahl and Haflidason, 1992). Hence, the younger tephra layers in Torfdalsvatn expand the Skógar Tephra's age uncertainty. In south Iceland, a lake sediment record from Hestvatn (Fig. 1a) contains four

bimodal Katla tephra layers above the assumed Vedde Ash, all of which are of similar composition to those in Torfdalsvatn and dated between ~11700 and 10600 cal a BP (Fig. 4b-c, Geirsdóttir et al., 2022). While there is greater uncertainty in Hestvatn's age model, it is possible that some of these tephra layers may correlate with those in Torfdalsvatn.

Similar to evidence from Iceland, records from elsewhere in the North Atlantic and Europe also indicate the presence of multiple Katla tephra layers with Vedde-like compositions during the Late Glacial and Early Holocene (e.g., Lane et al.,

2012). In Greenland ice cores (e.g., NGRIP), another rhyolitic tephra layer presumably from the Katla volcanic system is estimated to be <100 years older than the Vedde Ash (Mortensen et al., 2005; Cook et al., 2022), similar to evidence in the marine realm between Iceland and Greenland (Eiríksson et al., 2004; Gudmundsdóttir et al., 2012; Voelker and Haflidason, 2015). In south Sweden, two successive cryptotephra shard peaks associated with the Vedde Ash appear in lake sediment with [14]C ages of 12045 to 11975 cal a BP (Wastegård et al., 1998). In and around Scotland, the compositionally similar Abernethy

Tephra (Fig. 4d) is ~600 years younger than the Vedde Ash (Matthews et al., 2011; MacLeod et al., 2015; Muschitiello et al., 2019). In Lake Hämelsee, Germany, a cryptotephra, which possibly correlates to the Abernethy Tephra, as well as less concentrated glass shards of Vedde-like composition are also present above the Vedde Ash (Jones et al., 2018). Similarly,



tephra glass with Vedde composition found on the Lofoten Islands, Norway, is intermixed with the Askja S tephra layer, dated to $10830 \pm 57$ cal a BP (Pilcher et al., 2005; Bronk Ramsey et al., 2015).

160        Correlations between some of the Katla tephra layers in Torfdalsvatn and those in Hestvatn and marine sediment core MD99-2269 seem likely due to the similar composition, age, and northerly dispersal (Fig. 6). However, in addition to slightly different compositions, we suggest that similarly aged deposits in Europe (e.g., Abernethy Tephra) were generated during separate eruptions from those that produced the deposits found more locally in Iceland. Considering that these tephra deposits are rhyolitic and found distally in Europe, production from an explosive eruption and injection into the stratosphere is inevitable

(e.g., Thorarinsson, 1950; Sharma et al., 2008; Carey et al., 2010). Tephra dispersed at these atmospheric heights are generally carried eastward due to the prevailing westerly flowing jet stream. However, the winds are also seasonal, and during the spring/summer months, prevailing wind direction can shift to weak easterlies (Lacasse, 2001). The opposing directions of the north-western trajectory towards Torfdalsvatn (Fig. 6) and the eastern trajectory towards mainland Europe, as well as the short-lived nature of explosive rhyolitic eruptions (i.e., duration of individual events on the order of hours to days), makes it highly

unlikely that tephra layers found in both locations could have been generated from the same eruption.

       Given that tephra layers in general provide critical age control for many locations in the northern North Atlantic and Europe, improving ages of and correlations between proximal and distal tephra deposits is vital. The ~12000 cal a BP Vedde Ash has long stood as an important age control point for the Late Glacial period by providing constraint for ice sheet histories and sea level curves (e.g., Norðdahl and Haflidason, 1992; Rundgren et al., 1997; Farnsworth et al., 2022), local vegetation

records (Björck et al., 1992; Rundgren, 1995), the spatio-temporal evolution of abrupt North Atlantic climate change (Bakke et al., 2009; Lane et al., 2013), as well as insight into the timing of regional paleoceanography and marine reservoir ages (e.g., Koç et al., 1993; Eiríksson et al., 2000, 2004; Xiao et al., 2017; Muschitiello et al., 2019). However, the growing number of bimodal and silicic Katla tephra layers found during the Late Glacial to Early Holocene in Europe, and recently in Iceland, now requires detailed evaluations of prior correlations, especially if the records lack independent, supporting age control. As

the community moves forward, the identification of and correlation between these additional well-dated Vedde-like tephra layers will help bolster age models supporting paleoclimate records covering the rapid climate changes that occurred between the Late Glacial and Holocene. However, and echoing the conclusion of Lane et al. (2012), we stress that independent chronological constraint is imperative to make robust tephra layer correlations with the Vedde Ash, or otherwise, in any future studies.

**Data Availability**

All data is available in the Supplement.

**Author Contributions**

DJH and TT conceived the research; ÁG, GHM and CF acquired the lake sediment core; DJH and TT performed electron microscope analyses; DJH generated age models and wrote the manuscript with contributions from all co-authors.



**Competing Interests**

The authors declare they have no conflicts of interest.

**Acknowledgements**

This project has been principally supported by the Icelandic Center for Research (RANNÍS) through Grant-of-Excellences #022160002-04, #70272011-13 and #141573051-3 awarded to ÁG and co-authors as well as through the University of Iceland
Research Fund. DJH acknowledges support from RANNÍS Doctoral Student Grant #163431051 and CRF acknowledges support from the Doctoral Grant of the University of Iceland. We kindly thank Þorsteinn Jónsson and Sveinbjörn Steinþórsson for lake coring assistance.

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







**Table 1: Lake sediment radiocarbon information used in this study.**

| Lab ID | Depth (cm) | Material | Conventional $^{14}$C age $\pm$ $\sigma$ | Calibrated age BP $\pm$ $\sigma$ | Reference |
|--------|------------|----------|-------------------------------------------|-----------------------------------|-----------|
| Ua-1892 | 1035 | Bulk sediment | 8540 ± 230 | 9520 ± 370 | Björck et al. (1992) |
| Ua-1891 | 1052 | Bulk sediment | 8860 ± 250 | 9880 ± 320 | Björck et al. (1992) |
| Ua-1890 | 1060 | Moss macrofossil | 9180 ± 210 | 10330 ± 370 | Björck et al. (1992) |
| Ua-1889 | 1085 | Bulk sediment | 9890 ± 290 | 11340 ± 540 | Björck et al. (1992) |
| Ua-1888 | 1095 | Bulk sediment | 9470 ± 200 | 10800 ± 290 | Björck et al. (1992) |
| Ua-1887 | 1122 | Bulk sediment | 10550 ± 240 | 12400 ± 340 | Björck et al. (1992) |






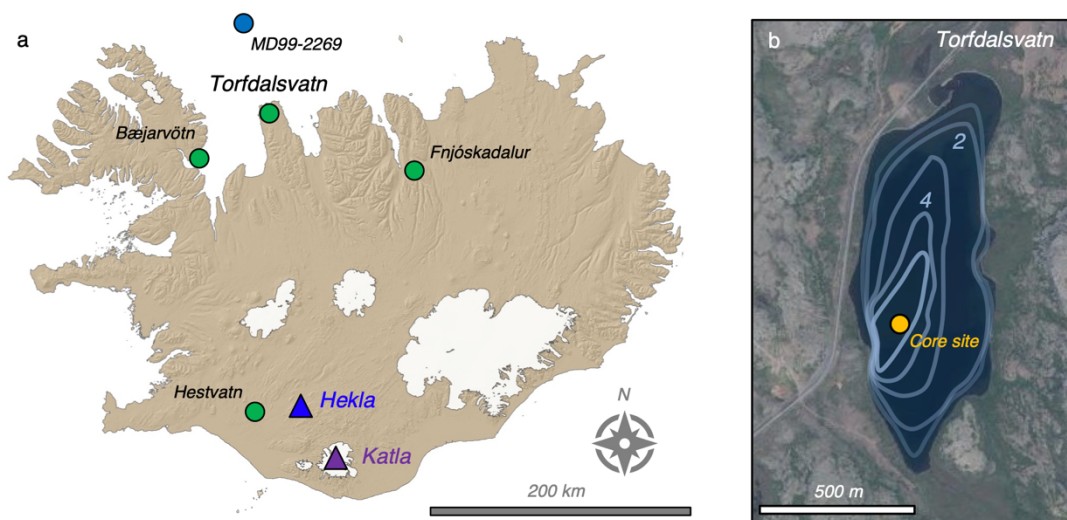

**Figure 1: Overview map of Iceland. A) Locations of Katla and Hekla central volcanos (triangles), and terrestrial (green) and marine sites (blue) mentioned in the text. B) Close-up of Torfdalsvatn, it's bathymetry (1-m contour lines), and location of lake sediment core site TORF12-1A-1B. Base image courtesy of Loftmyndir ehf.**

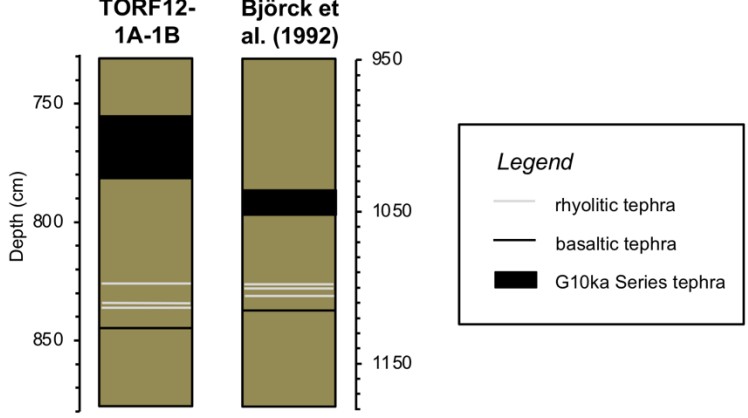

**Figure 2: Simplified stratigraphy of lake sediment core TORF12-1A-1B compared to that from Björck et al. (1992).**




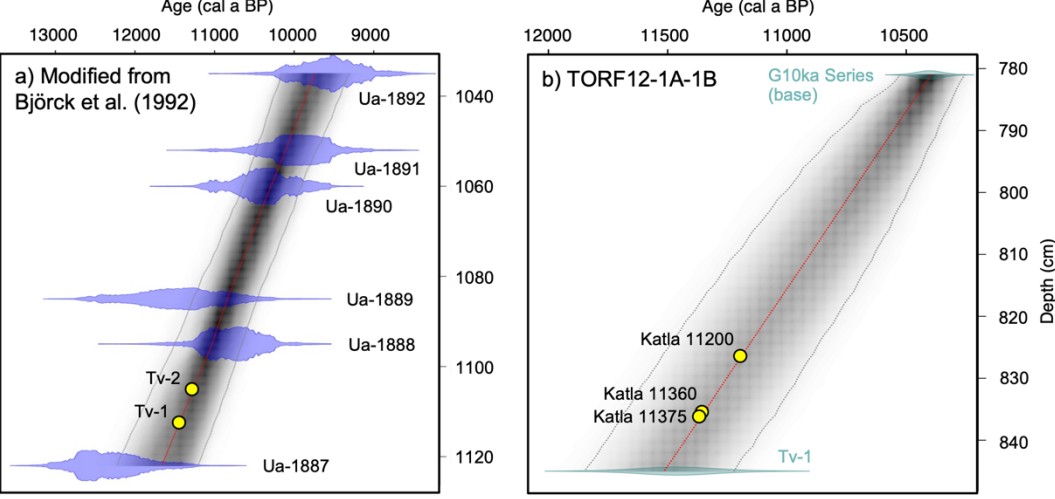

**Figure 3: Torfdalsvatn lake sediment age models. A) original sediment core from Björck et al. (1992) based on six** [14]**C ages (blue). B) TORF12-1A-1B based on two marker tephra layers (green). See Table 1 for complete radiocarbon information. Key tephra layers ages constrained from these models are marked in yellow. Red lines reflect mean values of model iterations, the gray lines denote the 95% confidence envelope, and darker shading reflects more likely ages.**






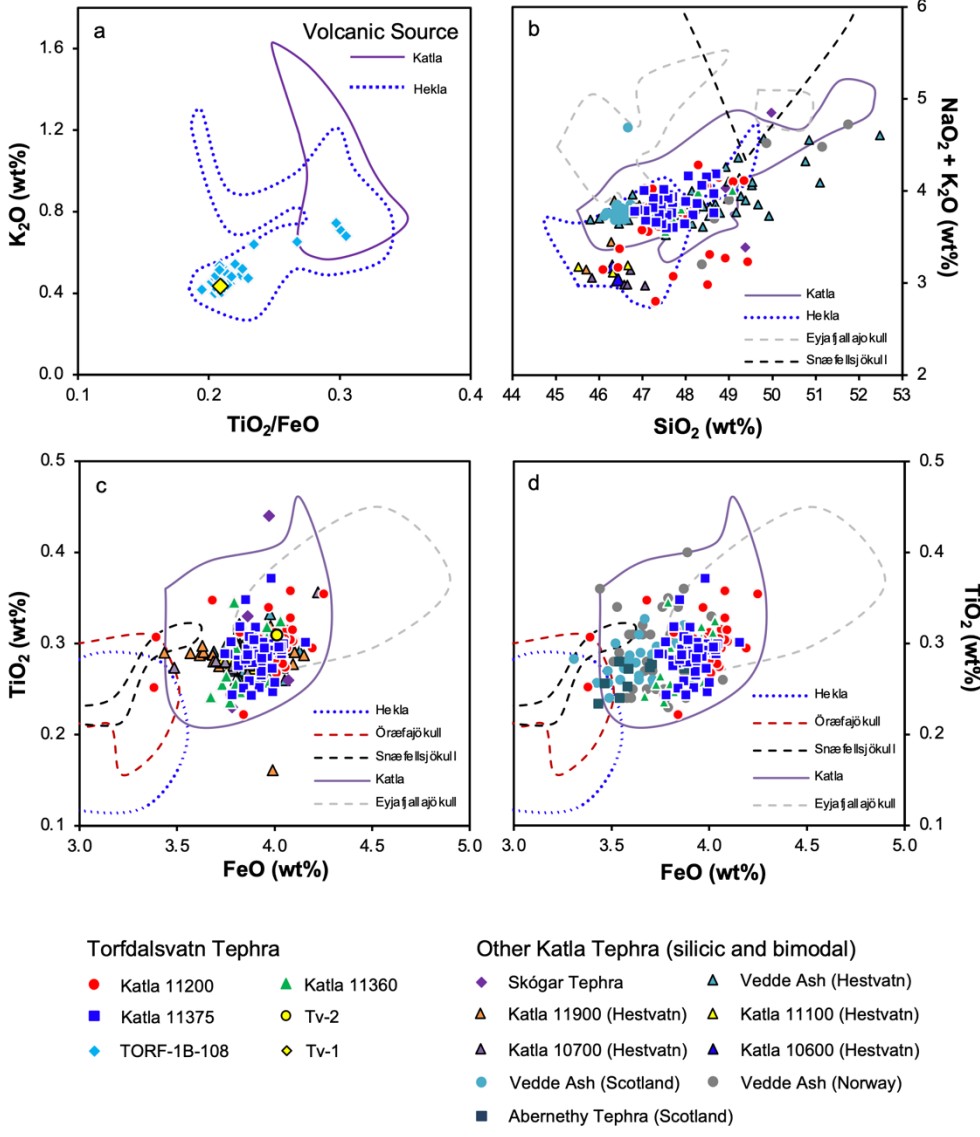

**Figure 4: Example source discrimination biplots. A) basaltic Hekla Tv-1 tephra, B) basaltic Katla tephra layers endmembers in Torfdalsvatn compared with Iceland and Europe, C) silicic Katla tephra layer endmembers in Torfdalsvatn compared with Iceland and C) silicic Katla tephra layer end-members in Torfdalsvatn compared with Europe. Tv-1 and Tv-2 geochemistries are average values (Björck et al., 1992). Please see the supporting data for raw data and further discrimination plots.**



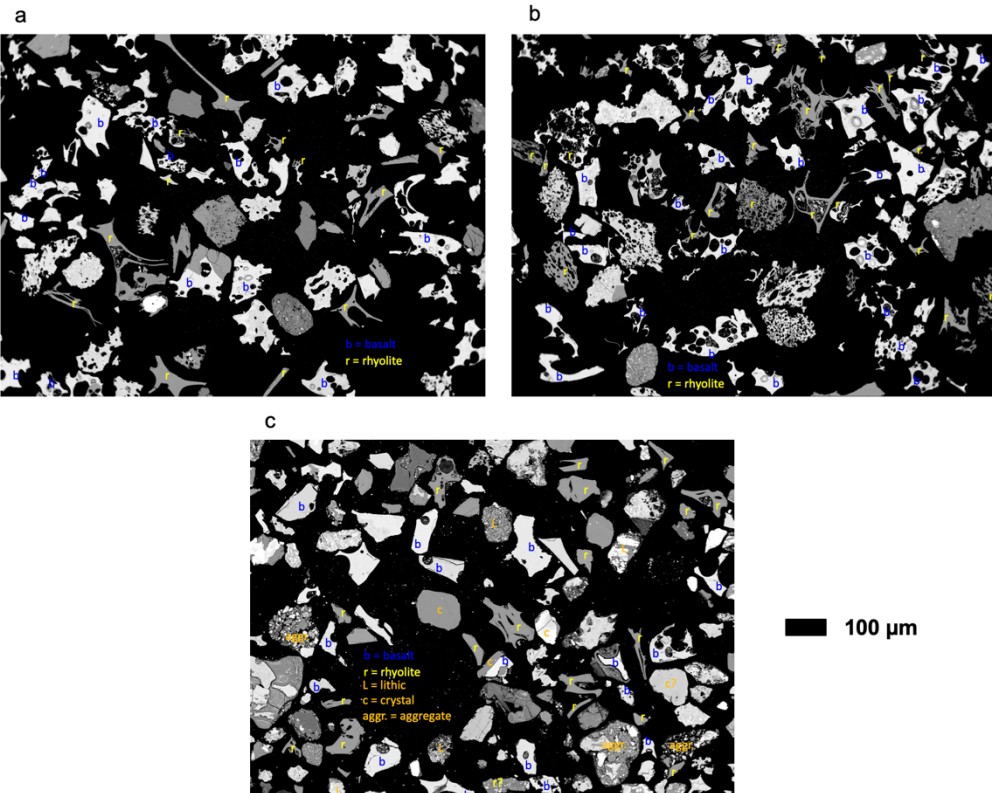

**Figure 5: Electron microprobe backscatter images for A) Katla 11375 BP (TORF-1B-100), B) Katla 11360 BP (TORF-1B-99), and C) Katla 11200 (TORF-1B-90).**






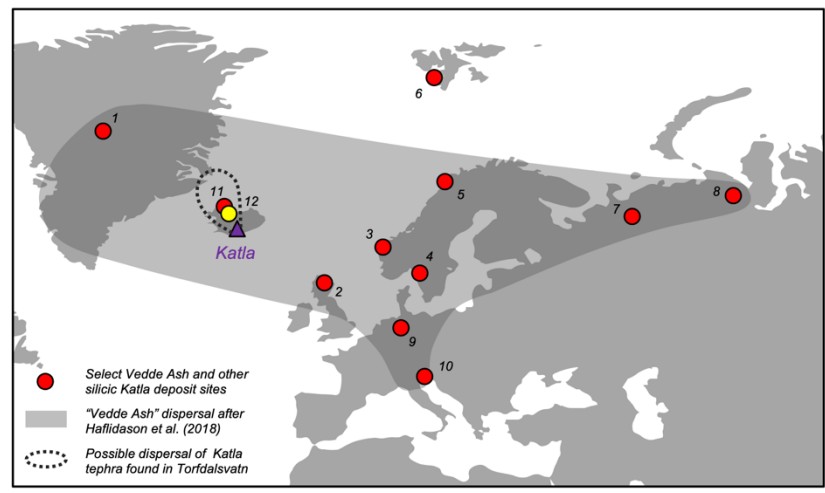

**Figure 6: Map of select sites with Vedde Ash and other similar silicic Katla deposits. 1) NGRIP ice core (Mortensen et al., 2005), 2)**
**Abernethy Forest, Scotland (Matthews et al., 2011; MacLeod et al., 2015), 3) Kråkenes, Norway (Mangerud et al., 1984), 4) Lake**
**Madtjärn, Sweden (Wastegård et al., 1998), 5) Lofoten Islands, Norway (Pilcher et al., 2005), Heftyevatnet (Farnsworth et al., 2022),**
**7) Lake Yamozero, Siberia (Haflidason et al., 2018), 8) Lake Bolshoye Shchuchye, Polar Ural Mountains, Siberia (Haflidason et al.,**
**2018), 9) Lake Hämelsee, Germany (Jones et al., 2018), 10) Lake Bled, Slovenia (Lane et al., 2011), 11) MD99-2269, North Iceland**
**Shelf (Kristjánsdóttir et al., 2007), and 12) Torfdalsvatn, Iceland. Gray shaded area is the most recent envelope for fallout deposits**
**of the presumed Vedde Ash (Haflidason et al., 2018).**