# Peer review of "Multiple Early Holocene eruptions of Katla produced tephra layers with similar composition to the Vedde Ash"

_Geochronology, 2022_

## Referee Comment (RC1)

[Figure]

**Fig 1:**

This graph shows an alternative age model for the Torfdalsvatn core based on ice-core ages of the Tv-1 and Tv-4 tephras (Mortensen et al. 2005; Cook et al. 2022). The NGRIP1519-1 layer, dated to c 12,646 b2k was correlated with Tv-1/I-THOL-2 by Mortensen et al. (2005). Biplots (Fig. 2) suggest that the correlation is correct, but Mortensen et al. tentatively sourced the layer to Grímsvötn, not Hekla/Vatnafjöll.

The age-depth line is drawn as linear interpolation between Tv-1 (I-THOL-2/NGRIP1519-1) and Tv-4 (G10-series/Saksunarvatn) with ages from NGRIP. This suggests that Tv-2 (Vedde?) in Torfdalsvatn has an age of c 12,350 b2k and Tv-3 (I-THOL-1) ca 10,900 b2k. This is a slightly high age for Vedde, but firmly within the Younger Dryas/GS-1, and not early Holocene.

**Fig. 2:**

The biplots on next page show that there is a good agreement between the Tv-1 tephra from Torfdalsvatn and the NGRIP1519.1/GRIP1654.05 from Greenland ice. There is more spread in the ice core samples which might be due to smaller shards that were analysed, but all major elements overlap.

[Figure]

[Figure]

**Fig. 3.**

Fig. 2 from Rundgren (1995) showing organic C, litho- and tephrostratigraphy, SIRM and pollen stratigraphy of the lowermost 1.05 m of sediments in L. Torfdalsvatn. The core is not the same that was investigated by Björck et al. (1992) but "was easily correlated with the previous one in the field on the basis of lithostratigraphy". Note the detection of redeposition in T-4 indicating a high sedimentation rate during this zone. The increase in organic content and pollen concentration values at the beginning of zone T-6 is interpreted as the transition from the Younger Dryas cold event into the Preboreal warming which is in agreement with the radiocarbon date of 9890 ± 290 [14]C y BP (Ua-1889).

If the interpretation of the pollen stratigraphy is correct, then the Vedde Ash (Tv-2) is firmly within the Younger Dryas sediments. Furthermore, Tv-1/I-THOL-2 is placed just below the onset of T-3 which marks a dramatic drop in pollen influx and organic carbon that points to a sudden change to a colder climate with a shorter ice-free season in the lake (Rundgren, 1995, p 411), presumably the Younger Dryas cooling. This is not compatible with an early Holocene age of 11460 cal a BP for the Tv-1 tephra

                                          MATS RUNDGREN

---

## Author Comment (AC1)

**Jan Mangerud**

We kindly thank Jan Mangerud for taking the time to consider and comment on our manuscript. Below we provide a response to each individual comment.

Very interesting manuscript. But:

- I would like to see counting of glass chards between and below the Vedde-like ash layers in order to better evaluate the possibility of redeposition. In western Norway we find a tail of redeposited Vedde ash well into the Holocene.

  We thank Dr. Mangerud for raising this point and allowing us to clarify our rationale for why these tephra layers are primary deposits. As outlined in the manuscript, our primary argument relies on the fact that all the layers contain pristine and non-abraded glass shards with the inclusion of minimal lithics, and that each tephra layer features sharp upper and lower contacts with the interstitial organic sediment.

  In addition to the above, it is also near impossible to get bulk reworking of a tephra layer as required by the presence of three discrete layers, taking place decades to centuries after its deposition, let alone two times in a row. Moreover, given that Torfdalsvatn's catchment is low relief with minimal topography, the bulk of the reworking would be wind derived, which primarily mobilizes sub-50-micron grains and hence the reworked tephra would be very fine ash and each storm input would be expected to be normally size graded due to settling through the water column (i.e., Stoke's Law).

  Finally, tephra grain counting between layers would not be useful to discern tephra redeposition in Iceland where tephra comprises the background. Given that it is a volcanic island, the parent material of all soils is volcanic (Arnalds, 2004). While glass shard counting can be useful for distal locations in Norway, where Icelandic tephra shards are either primary or secondary deposits, we always find various glass grains present in Icelandic lake sediment due to the constant mobilization of the soil into lakes from the surrounding catchment.

- I am surprised that they did not obtain new radiocarbon ages.

  We thank Dr. Mangerud for raising this point and allowing us to clarify our rationale for not obtaining new radiocarbon dates. The dates that we calibrated were indeed generated several decades ago, but there is no reason to believe that they are any less reliable than ones generated today. The Björck et al. (1992) samples were dated with high-precision AMS techniques that remain the primary method used. While the uncertainty can be larger in samples dated several decades ago, the median ages of the original dates and those today remain similar. As an example, we compare 14C ages from the original Torfdalsvatn study (Björck et al., 1992) and another from the mid-late 2000s (Axford et al., 2007) – see Table below. Both samples were taken near the base of

the G10ka Series tephra, and we recalibrated them using IntCal20 to make them directly comparable (Reimer et al., 2020). Given their similar stratigraphic location with respect the overlying tephra layers, the similar median age of the two samples is expected. The older sample from Björck et al. (1992) simply has a larger range of uncertainty, but the median age itself is not substantially different from a more recently dated sample (Axford et al., 2007).

In the revised manuscript, we will be sure to emphasize the higher uncertainty of our age estimates due to the old 14C dates, but that the median ages should be reliable. Ultimately, our results provide a baseline for future studies to improve age estimates and correlations to other localities.

| Lab ID | Depth below G10ka Series | Material | Conventional $^{14}$C age ± σ | Calibrated age BP ± σ | Reference |
|---|---|---|---|---|---|
| Ua-1890 | 8 cm | Moss macrofossil | 9180 ± 210 | 10330 ± 370 | Björck et al. (1992) |
| NSRL-14518 | 1.4 cm | Bulk sediment | 9100 ± 25 | 10240 ± 10 | Axford et al. (2007) |

References

Arnalds, O.: Volcanic soils of Iceland. Catena, 56, 3-20, https://doi.org/10.1016/j.catena.2003.10.002, 2004.

Axford, Y., Miller, G. H., Geirsdóttir, Á., and Langdon, P. G.: Holocene temperature history of northern Iceland inferred from subfossil midges. Quat. Sci. Rev., 26, 3344-3358, http://doi.org/10.1016/j.quascirev.2007.09.003, 2007.

Björck, S., Ingólfsson, Ó., Haflidason, H., Hallsdóttir, M., and Andersen, N. H.: Lake Torfadalsvatn: a high resolution record of the North Atlantic ash zone I and the last glacial-interglacial environmental changes in Iceland. Boreas, 21, 15-22, https://doi.org/10.1111/j.1502-3885.1992.tb00009.x, 1992.

Reimer, P. J., Austin, W. E. N., Bard, E., Bayliss, A., Blackwell, P. G., Bronk Ramsey, C., Butzin, M., Cheng, H., Edwards, R. L., Friedrich, M., Grootes, P. M., Guilderson, T. P., Hajdas, I., Heaton, T. J., Hogg, A. G., Hughen, K. A., Kromer, B., Manning, S. W., Muscheler, R., Palmer, J. G., Pearson, C., van der Plicht, J., Reimer, R. W., Richards, D. A., Scott, E. M., Southon, J. R., Turney, C. S. M., Wacker, L., Adolphi, F., Büntgen, U., Capano, M., Fahri, S. M., Fogtmann-Schulz, A., Friedrich, R., Köhler, P., Kudsk, S., Miyake, F., Olsen, J., Reinig, F., Sakamoto, M., Sookdeo, A., and Talamo, S.: The IntCal20 northern hemisphere radiocarbon age calibration curve (0-55 cal kBP). Radiocarbon, 62, 725-757, http://doi.org/10.1017/RDC/2020.41, 2020.

---

## Author Comment (AC2)

**Simon Larsson**

We kindly thank Simon Larsson for taking the time to consider and comment on our manuscript. Below we provide a response to each individual comment.

Tephra studies are becoming increasingly complex as we are refining methodology and analysis and thereby detecting hitherto unknown deposits, reinterpreting old ones, and having more and more trouble separating some of them (chemically and chronologically). It is good to see a study with the intention to highlight a possible problem with a well-used isochron such as the Vedde Ash, but there are some major issues with this manuscript that would probably have to be remedied for the conclusions to be accepted.

1. As already commented, the study is missing a complete tephrostratigraphic description and has only sampled visible tephra layers (I'm assuming - this is not explicitly stated. While these can be argued to be "stratigraphically separated", there is a high likelihood of redeposition in such an environment as that of the study site and a complete tephra count for the sequence is, in my opinion, required to motivate the layers as being separate and primary deposits.

   We thank Dr. Larsson for raising this point and allowing us to clarify our rationale for why these tephra layers are primary deposits. As outlined in the manuscript, our primary argument relies on the fact that all the layers contain pristine and non-abraded glass shards with the inclusion of minimal lithics, and that each tephra layer features sharp upper and lower contacts with the interstitial organic sediment.

   In addition to the above, it is also near impossible to get bulk reworking of a tephra layer as required by the presence of three discrete layers, taking place decades to centuries after its deposition, let alone two times in a row. Moreover, given that Torfdalsvatn's catchment is low relief with minimal topography, the bulk of the reworking would be wind derived, which primarily mobilizes sub-50-micron grains and hence the reworked tephra would be very fine ash and each storm input would be expected to be normally size graded due to settling through the water column (i.e., Stoke's Law).

   Finally, tephra grain counting between layers would not be useful to discern tephra redeposition in Iceland where tephra comprises the background. Given that it is a volcanic island, the parent material of all soils is volcanic (Arnalds, 2004). While glass shard counting can be useful for distal locations in Europe, where Icelandic tephra shards are either primary or secondary deposits, we always find various glass grains present in Icelandic lake sediment due to the constant mobilization of the soil into lakes from the surrounding catchment.

2. The study is missing a complete description of the creation of the age models. It is specified that it is Bayesian and that IntCal20 was applied to create the age model for

the older study's sediment sequence, but no other specifics and no reference for software used is included. The creation of the age model for the present study's sediment sequence is not described explicitly (should it be assumed that the same procedure is applied as for the older?).

We thank Dr. Larsson for raising this point and allowing us to clarify our construction of age models. In L66-76, we outline how we constructed age models for both sediment records, i.e., from Björck et al. (1992) and the one presented in this study (TORF12-1A-1B). L75-76 states that, "Age models for the Björck et al. (1992) and TORF12-1A-1B sediment records were both created using the R package rbacon and default settings (Fig. 3, Blaauw and Christen, 2011)." While we included a citation for the R package (rbacon), we realize that we had forgotten to include a citation for R itself, which will be included during revisions.

3. As already commented, the study re-uses quite old radiocarbon dates with wide error margins, recalibrating them to create an age model for the old sediment sequence. This is interesting for comparison purposes but should be interpreted with a great deal of care. The suggested ages for the three tephra layers are based on linear interpolation between two other tephra dates spaced >60 cm apart, which is in my opinion not too robust of an age model. The study would greatly benefit from, if not require, a more complete chronology.

We thank Dr. Larsson for raising this point and allowing us to clarify our rationale for not obtaining new radiocarbon dates. The dates that we calibrated were indeed generated several decades ago, but there is no reason to believe that they are any less reliable than ones generated today. The Björck et al. (1992) samples were dated with high-precision AMS techniques that remain the primary method used. While the uncertainty can be larger in samples dated several decades ago, the median ages of the original dates and those today remain similar. As an example, we compare 14C ages from the original Torfdalsvatn study (Björck et al., 1992) and another from the mid-late 2000s (Axford et al., 2007) – see Table below. Both samples were taken near the base of the G10ka Series tephra, and we recalibrated them using IntCal20 to make them directly comparable (Reimer et al., 2020). Given their similar stratigraphic location with respect the overlying tephra layers, the similar median age of the two samples is expected. The older sample from Björck et al. (1992) simply has a larger range of uncertainty, but the median age itself is not substantially different from a more recently dated sample (Axford et al., 2007).

In terms, of derived ages from our new sediment core, it is safe to assume that sedimentation rate is linear between the two layers as shown by multiple prior studies from the early portion of Torfdalsvatn's sediment record (Björck et al., 1992; Rundgren, 1995; Alsos et al., 2021). Therefore, we feel comfortable interpolating the ages between the two tephra age control points and deriving age estimates for the three bimodal Katla tephra layers.

*In the revised manuscript, we will be sure to emphasize the higher uncertainty of our age estimates due to the old 14C dates and age model interpolation, but that the median ages should be reliable. Ultimately, our results provide a baseline for future studies to improve age estimates and correlations to other localities.*

| Lab ID | Depth below G10ka Series | Material | Conventional $^{14}$C age ± σ | Calibrated age BP ± σ | Reference |
|---|---|---|---|---|---|
| Ua-1890 | 8 cm | Moss macrofossil | 9180 ± 210 | 10330 ± 370 | Björck et al. (1992) |
| NSRL-14518 | 1.4 cm | Bulk sediment | 9100 ± 25 | 10240 ± 10 | Axford et al. (2007) |

4. There is little-to-no lithostratigraphic description of the sample cores. This should be provided and expanded upon when comparing the sediment sequences of the older study and the current one, and hopefully this could provide better motivation for the assumptions being made about the tephra findings in the current study correlating to those in the older study.

*We thank Dr. Larsson for raising this point and agree that further descriptions of the core may be helpful. We will gladly include more detailed lithostratigraphic descriptions of our core TORF12-1A-1B in a revised manuscript and how it compares to that of Björck et al. (1992).*

References

Alsos, I. G., Lammers, Y., Kjellman, S. E., Merkel, M. K. F., Bender, E. M., Rouillard, A., Erlendsson, E., Gudmundsdóttir, E. R., Benediktsson, I. Ö., Farnsworth, W. F., Brynjólfsson, S., Gísladóttir, G., Eddudóttir, S. D., and Schomacker, A.: Ancient sedimentary DNA shows rapid post-glacial colonisation of Iceland followed by relatively stable vegetation until the Norse settlement (Landnám) AD 870. Quat. Sci. Rev., 259, 106903, https://doi.org/10.1016/j.quascirev.2021.106903, 2021.

Arnalds, O.: Volcanic soils of Iceland. Catena, 56, 3-20, https://doi.org/10.1016/j.catena.2003.10.002, 2004.

Axford, Y., Miller, G. H., Geirsdóttir, Á., and Langdon, P. G.: Holocene temperature history of northern Iceland inferred from subfossil midges. Quat. Sci. Rev., 26, 3344-3358, http://doi.org/10.1016/j.quascirev.2007.09.003, 2007.

Björck, S., Ingólfsson, Ó., Haflidason, H., Hallsdóttir, M., and Andersen, N. H.: Lake Torfadalsvatn: a high resolution record of the North Atlantic ash zone I and the last glacialinterglacial environmental changes in Iceland. Boreas, 21, 15-22, https://doi.org/10.1111/j.1502-3885.1992.tb00009.x, 1992.

Blaauw, M., and Christen, J. A.: Flexible paleoclimate age-depth models using an autoregressive gamma process. Bayesian Anal., 6, 457–474, http://doi.org/10.1214/11-BA618, 2011.

Reimer, P. J., Austin, W. E. N., Bard, E., Bayliss, A., Blackwell, P. G., Bronk Ramsey, C., Butzin, M., Cheng, H., Edwards, R. L., Friedrich, M., Grootes, P. M., Guilderson, T. P., Hajdas, I., Heaton, T. J., Hogg, A. G., Hughen, K. A., Kromer, B., Manning, S. W., Muscheler, R., Palmer, J. G., Pearson, C., van der Plicht, J., Reimer, R. W., Richards, D. A., Scott, E. M., Southon, J. R., Turney, C. S. M., Wacker, L., Adolphi, F., Büntgen, U., Capano, M., Fahri, S. M., Fogtmann-Schulz, A., Friedrich, R., Köhler, P., Kudsk, S., Miyake, F., Olsen, J., Reinig, F., Sakamoto, M., Sookdeo, A., and Talamo, S.: The IntCal20 northern hemisphere radiocarbon age calibration curve (0-55 cal kBP). Radiocarbon, 62, 725-757, http://doi.org/10.1017/RDC/2020.41, 2020.

Rundgren, M.: Biostratigraphic evidence of the Allerød-Younger Dryas-Preboreal Oscillation in Northern Iceland. Quat. Res., 44, 405-416, https://doi.org/10.1006/qres.1995.1085, 1995.

---

## Author Comment (AC3)

I recently reviewed a similar manuscript by the same authors for another journal. I had raised several critical comments, and based on my review, the editor decided to reject the ms. I was happy to see that the authors decided to submit the ms again to GChron, but my main criticism remains after reading through the text. I summarise my main critical comments here. See also graphs in the supplement.

We kindly thank Referee 1 for their time considering and reviewing our submitted manuscript. We would like to highlight that yes, while the manuscript was previously submitted to another journal and rejected, the two reviews received reached opposing conclusions and recommendations. The other reviewer found our manuscript to provide a rigorous reassessment of previously published datasets and recommended only small clarifications, and the editor provided no independent assessment. Ultimately, it was unclear why one review was weighed on for the final decision.

Below we provide a response to each individual comment from Referee 1.

1) It is feasible that several eruptions of Katla occurred in the early Holocene, producing Vedde-type tephras. The results presented here, however, do not support this conclusion. The age-depth model is based on radiocarbon dates performed more than 30 years ago with standard deviations of several hundred years. Age-depth modelling of such old dates cannot give reliable ages and new samples for radiocarbon dating should have been submitted. The second lowest date (Ua-1888) is potentially an outlier, and by removing it from the model, all depths would get older ages in line with the interpretation of Björck et al. (1992)

We thank the reviewer for raising this point and allowing us to clarify our rationale. The dates that we calibrated were indeed generated several decades ago, but there is no reason to believe that they are any less reliable than ones generated today. The Björck et al. (1992) samples were dated with high-precision AMS techniques that remain the primary method used. While the uncertainty can be larger in samples dated several decades ago, the median ages of the original dates and those today remain similar. As an example, we compare 14C ages from the original Torfdalsvatn study (Björck et al., 1992) and another from the mid-late 2000s (Axford et al., 2007) – see Table below. Both samples were taken near the base of the G10ka Series tephra, and we recalibrated them using IntCal20 to make them directly comparable (Reimer et al., 2020). Given their similar stratigraphic location with respect the overlying tephra layers, the similar median age of the two samples is expected. The older sample from Björck et al. (1992) simply has a larger range of uncertainty, but the median age itself is not substantially different from a more recently dated sample (Axford et al., 2007).

Regarding the second lowest date (Ua-1888), there is no reason to believe that this sample is an outlier. First, evidence from Iceland shows that bulk sediment 14C outliers are typically stratigraphically too old due to the transport of terrestrial-derived carbon to the lake sediment (e.g., Geirsdóttir et al., 2009). However, the strong agreement between bulk sediment and macrofossil 14C ages from Axford et al. (2007) and Björck et al. (1992) (see Table below), respectively, suggest that these Early Holocene bulk sediment 14C ages are relatively accurate.

Moreover, given that this is a recently deglaciated landscape that likely has minimal terrestrial contributions due to a lack of soil formation, stratigraphically older ages from bulk sediment are unlikely. Second, in terms of a bulk sediment 14C age being stratigraphically too young, to the best of our knowledge, there is no viable mechanism that can naturally produce this scenario. In this sense, the age model should be weighted more towards younger ages as reflected in Fig. 3 for these records. Therefore, we believe that our interpretation of the data and derived age model remains valid.

We will expand our discussion in the manuscript (L79-86) to include the comparison of 14C ages published in 1992 and 2007 (Björck et al., 1992; Axford et al., 2007). Moreover, we will emphasize the higher uncertainty of our age estimates due to the old 14C dates, but that the median ages should be reliable. Ultimately, our results provide a baseline for future studies to improve age estimates and correlations to other localities. We hope this allays any concerns of unreliable ages.

| Lab ID | Depth below G10ka Series | Material | Conventional $^{14}$C age ± σ | Calibrated age BP ± σ | Reference |
|---|---|---|---|---|---|
| Ua-1890 | 8 cm | Moss macrofossil | 9180 ± 210 | 10330 ± 370 | Björck et al. (1992) |
| NSRL-14518 | 1.4 cm | Bulk sediment | 9100 ± 25 | 10240 ± 10 | Axford et al. (2007) |

2) There is no discussion about the possibility of reworking of Vedde shards in the early Holocene. There are many examples from western Norway of reworking many thousand years into the Holocene. I agree with Mangerud's comments that a tephra count graph would be helpful here.

We thank the reviewer for raising this point and allowing us to clarify our rationale. We respectfully refer the reviewer to both the Section 3.2 (Tephra layer descriptions, L98-120) and Discussion (L122-130) sections where we discuss why reworking of these tephra layers is improbable. Our primary argument relies on the fact that all the layers contain pristine and non-abraded glass shards with the inclusion of minimal lithics and that each tephra layer features sharp upper and lower contacts with the interstitial organic sediment.

In addition to the above, it is also near impossible to get bulk reworking of a tephra layer as required by the presence of three discrete layers, taking place decades to centuries after its deposition, let alone two times in a row. Moreover, given that Torfdalsvatn's catchment is low relief with minimal topography, the bulk of the reworking would be wind derived, which primarily mobilizes sub-50-micron grains and hence the reworked tephra would be very fine ash and each storm input is expected to be normally size graded due to settling through the water column (i.e., Stoke's Law).

Finally, tephra grain counting between layers would not be useful to discern tephra redeposition in Iceland where tephra comprises the background. Given that it is a volcanic

island, the parent material of all soils is volcanic (Arnalds, 2004). While glass shard counting can be useful for distal locations like Norway, where Icelandic tephra shards are either primary or secondary deposits, we always find various glass grains present in Icelandic lake sediment due to the constant mobilization of the soil into lakes from the surrounding catchment.

3) The lowest tephra layer, Tv-1 has a Hekla-like geochemistry and is believed to be one of the oldest basaltic layers from Hekla. The authors, however, do not mention that Tv-1 has been correlated with a tephra found in the NGRIP ice-core, NGRIP1519-1, dated to c 12,646 b2k (Mortensen et al., 2005; JQS). New data from NGRIP and GRIP confirms this correlation and firmly places the Tv-1/NGRIP1519-1 in the early part of Younger Dryas/GS-1 (Cook et al., 2022; QSR). The attached graph (Fig. 1 in the supplement) shows an alternative age model for the Torfdalsvatn core based on ice-core ages of the Tv-1 and Tv-4 tephras (Mortensen et al. 2005; Cook et al. 2022). It suggests that the Tv-2 layer (Vedde in Björck et al's paper) is firmly placed in the YD and not in early Preboreal. Biplots (Fig. 2 in the supplement) show that there is a generally good agreement between the Tv-1 tephra from Torfdalsvatn and the NGRIP1519.1/GRIP1654.05 layer from Greenland ice. There is more spread in the ice core samples which might be due to smaller shard that were analysed, but all major elements overlap.

We thank the reviewer for raising this point and allowing us to clarify our rationale. As previously outlined by studies, some which use the Vedde Ash as an example (e.g., Lane et al., 2012), chemistry alone is not sufficient to draw a tephra layer correlation. Basaltic and rhyolitic tephra layers from Iceland are known to share indistinguishable chemistries during the Holocene (e.g., Larsen et al., 2001; Óladóttir et al., 2020), meaning that independent age control is imperative to identify the tephra layer and derive secure correlations between geographic regions. While the reviewer is correct that the tephra in the Greenland ice cores shares similar major oxide geochemistry with the Tv-1 tephra layer in Torfdalsvatn, the objective age constraint we provide by calibrating the Björck et al. (1992) age model indicates that the Tv-1 tephra layer in Torfdalsvatn is younger and therefore must be from different volcanic eruption.

As noted by the reviewer, prior studies have suggested correlations between additional Torfdalsvatn and Greenland tephra layers. However, these correlations relied on the old ages estimates from Björck et al. (1992), which itself partially relied on the assumption that Tv-2 was the Vedde Ash (~12100 cal BP). The alternative age model that the reviewer provides uses the ages from the Greenland ice core for Torfdalsvatn's tephra layers based on the assumption that the original Torfdalsvatn age model was correct (e.g., Björck et al., 1992). We respectfully argue that using this alternative age model to justify older ages of Torfdalsvatn's Tv-1 tephra layer is circular reasoning. In this regard, we favor our objective approach that independently dates Torfdalsvatn's tephra layers with quantitative dating techniques from the same record. While certainly not confirming the age of Torfdalsvatn's tephra layers, there are contemporaneous tephra layers of similar composition on the North Iceland Shelf (Kristjánsdóttir et al., 2007) and

in Hestvatn (Geirsdóttir et al., 2021), which provide possible correlations, as discussed in the manuscript (L131-147).

Given our above points, and our response to issue 1 regarding the reliability of our revised age model, we believe that our interpretation of tephra layer ages in Torfdalsvatn remains valid. We will expand upon our description of the Tv-1 tephra and its possible correlations to other North Atlantic sites in the manuscript.

4) Previous investigations at Torfdalsvatn by Rundgren (1995, QR) are mentioned in the text, but rather surprisingly, are not discussed in any detail. The pollen- and lithostratigraphy presented by Rundgren suggest an YD age for Tv-2 and an early YD age for Tv-1, in agreement Björck's paper, see also Fig. 3 in the supplement file.

We thank the reviewer for raising this point and allowing us to clarify our rationale for not discussing the published pollen datasets (Björck et al., 1992; Rundgren, 1995). First, we want to emphasize that the Younger Dryas Stadial (12900 to 11700 cal a BP) cannot be identified based on pollen and lithostratigraphy without independent chronological control. A relatively recent study emphasized this issue well through high-resolution dating of two terrestrial archives to demonstrate the times-trangressive nature of Younger Dryas-associated cooling in Europe (Lane et al., 2013). If these two records were dated according to their proxy records, such as biostratigraphy, they would both cover the same age range, which we now believe is not the case. Given that we calibrate the original Björck et al. (1992) age model, which was then adopted by Rundgren (1995) as highlighted by the reviewer, the revised age constraint suggests that the pollen-inferred cooling post-dates the Younger Dryas. We prefer to not include further discussion of the pollen spectra in this paper as the vegetation history does not relate to our study's focus, which is the evaluation of the Torfdalsvatn's Early Holocene 14C ages and tephra layers.

References

Arnalds, O.: Volcanic soils of Iceland. Catena, 56, 3-20, https://doi.org/10.1016/j.catena.2003.10.002, 2004.

Axford, Y., Miller, G. H., Geirsdóttir, Á., and Langdon, P. G.: Holocene temperature history of northern Iceland inferred from subfossil midges. Quat. Sci. Rev., 26, 3344-3358, http://doi.org/10.1016/j.quascirev.2007.09.003, 2007.

Björck, S., Ingólfsson, Ó., Haflidason, H., Hallsdóttir, M., and Andersen, N. H.: Lake Torfadalsvatn: a high resolution record of the North Atlantic ash zone I and the last glacial-interglacial environmental changes in Iceland. Boreas, 21, 15-22, https://doi.org/10.1111/j.1502-3885.1992.tb00009.x, 1992.

Geirsdóttir, Á., Miller, G. H., Harning, D. J., Hannesdóttir, H., Thordarson, T., and Jónsdóttir, I.: Recurrent outburst floods and explosive volcanism during the Younger Dryas-Early Holocene

deglaciation in south Iceland: evidence from a lacustrine record. J. Quat. Res., 37, 1006-1023, https://doi.org/10.1002/jqs.3344, 2022.

Geirsdóttir, Á., Miller, G. H., Thordarson, T., and Ólafsdóttir, K. B.: A 2000 year record of climate variations reconstructed from Haukadalsvatn, West Iceland. J. Paleolimnol., 41, 95-115, https://doi.org/10.1007/s10933-008-9253-z, 2009.

Kristjánsdóttir, G. B., Stoner, J. S., Jennings, A. E., Andrews, J. T., and Grönvold, K.: Geochemistry of Holocene cryptotephras from the North Iceland Shelf (MD99-2269): intercalibration with radiocarbon and palaeomagnetic chronostratigraphies. Holocene, 17, 155-176, https://doi.org/10.1177/0959683607075829, 2007.

Lane, C. S., Blockley, S. P. E., Mangerud, J., Smith, V. C., Lohne, Ø. S., Tomlinson, E. L., Matthews, I. P., and Lotter, A. F.: Was the 12.1 ka Icelandic Vedde Ash one of a kind? Quat. Sci. Rev., 33, 87-99, https://doi.org/10.1016/j.quascirev.2011.11.011, 2012.

Lane, C. S., Brauer, A., Blockley, S. P. E., and Dulski, P.: Volcanic ash reveals time-transgressive abrupt climate change during the Younger Dryas. Geology, 41, 1251-1254, https://doi.org/10.1130/G34867.1, 2013.

Larsen, G., Newton, A. J., Dugmore, A. J., Vilmundardóttir, E. G.: Geochemistry, dispersal, volumes and chronology of Holocene silicic tephra layers from the Katla volcanic system, Iceland. J. Quat. Sci., 16, 119-132, https://doi.org/10.1002/jqs.587, 2001.

Óladóttir, B. A., Thordarson, T., Geirsdóttir, Á., Jóhannsdóttir, G. E., and Mangerud, J.: The Saksunarvatn Ash and the G10ka series tephra. Review and current state of knowledge. Quat. Geochron., 56, 101041, http://doi.org/10.1016/j.quageo.2019.101041, 2020.

Reimer, P. J., Austin, W. E. N., Bard, E., Bayliss, A., Blackwell, P. G., Bronk Ramsey, C., Butzin, M., Cheng, H., Edwards, R. L., Friedrich, M., Grootes, P. M., Guilderson, T. P., Hajdas, I., Heaton, T. J., Hogg, A. G., Hughen, K. A., Kromer, B., Manning, S. W., Muscheler, R., Palmer, J. G., Pearson, C., van der Plicht, J., Reimer, R. W., Richards, D. A., Scott, E. M., Southon, J. R., Turney, C. S. M., Wacker, L., Adolphi, F., Büntgen, U., Capano, M., Fahri, S. M., Fogtmann-Schulz, A., Friedrich, R., Köhler, P., Kudsk, S., Miyake, F., Olsen, J., Reinig, F., Sakamoto, M., Sookdeo, A., and Talamo, S.: The IntCal20 northern hemisphere radiocarbon age calibration curve (0-55 cal kBP). Radiocarbon, 62, 725-757, http://doi.org/10.1017/RDC/2020.41, 2020.

Rundgren, M.: Biostratigraphic evidence of the Allerød-Younger Dryas-Preboreal Oscillation in Northern Iceland. Quat. Res., 44, 405-416, https://doi.org/10.1006/qres.1995.1085, 1995.

---

## Author Comment (AC4)

We kindly thank Referee 2 for their time considering and reviewing our submitted manuscript. Below we provide a response to each of the two comments from Referee 2.

Ultimately, sharp contacts and pristine glass shards do not provide robust evidence that an ash layer is primary. Moreover, since the tephra layers identified here are compositionally identical and in sediments younger than the Vedde ash, other lines of evidences are essential to categorically prove these are not redeposited Vedde ash. Therefore I find the conclusions of the paper here are unsupported. Numerous other high-resolution lake records have shown that it is extremely difficult to discriminate reworked ash from older (primary) events, and that ash can be redeposited, producing discrete visible and cryptic layers with perfect pristine glass shards (even over tens of thousands of years after the original eruption). Rigorous morphological, geochemical and sedimentological work has not been demonstrated, and may not be able to rule out the possibility of redeposition.

We thank the reviewer for raising this point and allowing us to clarify our rationale.

The claim put forward by Reviewer 2 that: „…., sharp contacts and pristine glass shards do not provide robust evidence that an ash layer is primary" is misguided and, as presented, unsupported by any evidence or observations. This claim is also somewhat surprising, because both observations and measurements have demonstrated that grain morphology/grain shape is perhaps one of the best tools available to differentiate between primary and reworked deposits (e.g., Wilcox and Naeser, 1992; Leahy, 1997; Guðmundsdóttir et al 2011; Lowe, 2011; Óladóttir et al 2011; Dugmore et al., 2020; see also quotes extracted from these publications below). Additional aspects that help to define primary tephra layers are 1) homogenous geochemical compositions, 2) minimal incorporation of exotic material (e.g., lithic fragments, biological microfossils, etc.), 3) lack of sedimentary structures such as turbidites that would indicate redeposition, and 4) spatial distribution of the layer across different landscapes and deposits (Lacasse et al., 1998; Boygle, 1999, Shane et al., 2006; Gudmundsdóttir et al., 2011; Lowe, 2011; Dugmore et al., 2020):

**Quote from Guðmundsdóttir et al 2011**
"Morphological measurements and microprobe analyses were used to discriminate between primary and reworked tephra. The morphological measurements reveal fresh glass grains at the intervals where the nine tephra layers have been located, demonstrating the usefulness of this method to evaluate whether a tephra layer is primary or reworked. It appears that a ruggedness value of less than 0.4 is indicative for primary tephra in this environment. Microprobe analyses are unsurpassed as a tool to correlate tephra to source volcanic system and to distinguish between a primary tephra layer, which manifests itself by a dominant glass composition, and reworked tephra consisting of grains with several different glass compositions."

**Quote from Óladóttir et al 2011**
„When defining primary tephra both field observations and chemical composition are important. Three field observations have proved useful in distinguishing between primary

tephra and reworked material: (1) Colour and contacts. Primary tephra has distinct colour as it is not contaminated by soil and unrelated tephra grains. Sharp contacts indicate primary undisturbed tephra. Laterally continuous bedding can also be diagnostic. (2) Grain size and shape. During wind erosion grains become abraded and sorted, leading to decreasing grain size and increased sorting in reworked material compared with primary tephra. (3) Thickness. Thick tephra can cover vegetation completely, increasing the possibilities of tephra reworking. A homogeneous chemical composition confirms the primary character of tephra whereas heterogeneous results may indicate (1) contamination from surrounding units, (2) contemporaneous eruptions at two or more volcanic systems or (3) reworked material."

**Quote from Lowe (2011) – tephrochronology review paper**
"Various other laboratory analyses can also provide clues that tephra reworking has occurred. For example, partial rounding of grains in a tephra layer and the loss of glassy coatings from fresh crystals both point towards a reworking event (e.g., Wilcox and Naeser, 1992; Leahy, 1997). If the major element concentrations of glass shards in a tephra are normally homogeneous (i.e., analyses of individual shards show only small deviations from one shard to the next), then multiple populations of such shards indicate that post-depositional mixing has probably occurred, or that two or more tephras were deposited effectively simultaneously from closely-spaced eruptions."

With these criteria for defining primary tephra layers in mind, we respectfully disagree with the reviewer that our evidence for primary deposits is unsupported. All tephra layers described in our study have sharp contacts, pristine shard morphometry, tight geochemical populations, do not contain any substantial exotic material, and feature large grain sizes that cannot be mobilized by wind and are not normally graded as a redeposited turbidite would be. The tephra layers also have potential correlations in both a south Iceland lake and the marine realm north of Iceland (see main manuscript). Moreover, Torfdalsvatn's lake catchment is low relief, which makes remobilization of soil and sediment challenging. Finally, the stratigraphical replication of Björck et al.'s 1992 record with our team's newer 2012 core further supports the presence of multiple primary tephra deposits, a principle previously highlighted specifically for Icelandic lake sediments (Boygle, 1999).

While the reviewer is correct that ash can be reworked as layers or as cryptotephra (Boylge, 1999; Wutke et al., 2015), our collective lines of evidence, all of which are supported by the literature, strongly suggest the tephra layers we evaluated are primary deposits. We recognize that some of the arguments supporting this conclusion may not have been clearly articulated in the submitted manuscript. Therefore, we will revise the manuscript to include a Discussion section dedicated to clearly outlining our reasoning for identifying the tephra layers as primary deposits. We hope that this will allay any concerns of redeposition by the reviewer.

The authors also need to show the integrity of the lake sediments, showing the tephrostratigraphy over a longer period of sedimentation and show the compositions of glass shards incorporated within the sediments over different timescales. I would suggest presenting the complete tephrostratigraphy of the lake sediments and including the work in this manuscript alongside a discussion of interpreting taphonomic issues like these that are commonly faced by the tephrochronological community working with sedimentary records.

We thank the reviewer for raising this point and allowing us to clarify our rationale. While the comment is unfortunately somewhat unclear to us, a discussion of tephra shards from non-primary tephra layers (i.e., background tephra) seems irrelevant to this study. Tephra preservation is certainly an open research direction in the field, however, our aim here is to report on primary ash deposits that may serve as regional age control points. As we have detailed above, our collective evidence strong supports that the tephra layers in this study are primary airfall deposits. We will keep the reviewer's suggestion to explore taphonomic processes of tephra in the background sediment in mind as a potential avenue for future research, but we would like to stress that this information is not relevant for the conclusions drawn in our current study.

References

Boygle, J.: Variability of tephra in lake and catchment sediments, Svínavatn, Iceland, Glob. Planet. Change, 21, 129-149, 1999.

Dugmore, A. J., Thompson, P. J., Streeter, R. T., Cutler, N. A., Newton, A. J., and Kirkbride, M. P.: The interpretative value of transformed tephra sequences, J. Quat. Sci., 35, 23-38, 2020.

Gudmundsdóttir, E. R., Eiríksson, J., and Larsen, G.: Identification and definition of primary and reworked tephra in Late Glacial and Holocene marine shelf sediments off North Iceland, J. Quat. Sci., 26, 589-602, 2011.

Lacasse, C., Werner, R., Paterne, M., Sigurdson, H., Carey, S., and Pinte, G.: Long-range transport of Icelandic tephra to the Irminger Basin, Site 919. In: Saunders, A.D., Larsen, H.C., Wise Jr., S.W. (Eds.), Proceedings of the Ocean Drilling Program. Scientific Results, vol. 152, pp. 51-65, 1998.

Leahy, K.: Discrimination of reworked pyroclastics from primary tephra-fall tuffs: a case study using kimberlites of Fort a la Corne, Saskatchewan, Canada, Bull. Volcanol., 59, 65-71, 1997.

Lowe, D. J.: Tephrochronology and its application: A review, Quat. Geochron., 6, 107-153, 2011.

Óladóttir, B. A., Sigmarsson, O., Larsen, G., and Devidal, J.-L.: Provenance of basaltic tephra from Vatnajökull subglacial volcanoes, Iceland, as determined by major- and trace-element analyses, Holocene 21, 1037–1048, 2011.

Shane, P. A. R., Sikes, E. L., and Guilderson, T. P.: Tephra beds in deep-sea cores off northern New Zealand: implications for the history of Taupo volcanic zone, Mayor Island and White Island volcanoes. J. Volcanol. Geotherm. Res., 154, 276-290, 2006.

Wilcox, R. E., and Naeser, C. W.: The Pearlette family ash beds in the Great Plains: finding their identities and their roots in the Yellowstone country, Quat. Int., 13-14, 9-13, 1992.

Wutke, K., Wulf, S., Tomlinson, E. L., Hardiman, M., Dulski, P., Luterbacher, J., and Brauer, A.: Geochemical properties and environmental impacts of seven Campanian tephra layers deposited between 40 and 38 ka BP in the varved lake sediments of Lago Grande di Monticchio, southern Italy, Quat. Sci. Rev., 118, 67-83, 2015.